# Hyaluronic Acid-Based Medical Device for Treatment of Alveolar Osteitis—Clinical Study [note 1]

**DOI:** 10.3390/ijerph16193698

**Published:** 2019-10-01

**Authors:** Jakub Suchánek, Romana Koberová Ivančaková, Radovan Mottl, Klára Zoe Browne, Kristýna Charlotte Pilneyová, Nela Pilbauerová, Jan Schmidt, Tereza Suchánková Kleplová

**Affiliations:** 1Department of Dentistry, Charles University – Faculty of Medicine in Hradec Králové and University Hospital Hradec Králové, Sokolská 581, 500 05 Hradec Králové, Czech Republic; 2Department of Histology and Embryology, Charles University – Faculty of Medicine in Hradec Králové, Šimkova 870, 500 03 Hradec Králové, Czech Republic

**Keywords:** alveolitis sicca dolorosa, alveolar osteitis, alveolalgia, sodium hyaluronate, octenidine dihydrochloride, clinical study, first-in-man study

## Abstract

Alveolar Osteitis (AO) is a complication following the extraction of a tooth. AO manifests through localized pain in, and around, the extraction site, where the post-operative blood clot has been disintegrated. The aim of this single cohort study was to evaluate the outcome of a treatment of AO, using a pharmacological device composed of hyaluronic acid and octenidine dihydrochloride. The tested device is a sponge-like material, composed solely of a fully dissoluble medicaments (hyaluronic acid, calcium chloride, and octenidine dihydrochloride). It was designed to serve as a non-toxic, slow-dissolving antiseptic, that adheres to mucosa and obturates the wound. This study includes 58 subjects who were diagnosed with AO. The tested device was administered once daily until local pain subsided to < 20 mm of the Visual Analog Scale (VAS). The treatment was considered effective when the pain subsided to < 20 mm VAS in < 8 days of treatment; as per comparative studies. Our findings provide a statistically significant success rate of 96.0% (95.0% confidence interval of 75.75% to 97.8%) after pharmacological device administrations. No adverse medical effects were detected. Acquired data confirmed that lyophilized hyaluronic acid, combined with octenidine, is effective for the treatment of AO. The results are clinically important as AO is a common complication after third molar extractions.

## 1. Introduction

Alveolar Osteitis (AO), also known as a dry socket, is a complication following the extraction of a tooth. It generally sets off within three days of the surgical intervention. The first symptom is severe pain in, and around, the extraction wound, irradiating into the jaw or temporal area, halitosis, and possible trismus [1]. The etiology of AO is frequently attributed to partial or total disintegration of a post-extraction blood clot and endogenous fibrinolysis. Among the most common causes and risk factors of AO are trauma or complicated extraction, smoking, or acute inflammation of periodontal tissues (e.g., dentitio difficilis) prior to extraction. Also, the use of oral contraception leads to a 10 times higher likelihood of AO [2,3,4,5,6,7]. Furthermore, secondary contributing risk factors, include age, flap design, local anesthetic containing vasoconstrictor, and bone/root debris left in the extraction wound [7]. Overall, according to Field et al. [8], MacGregor [9], Turner [10], and Krogh [11], the frequency of AO varies from 0.5% to 5.0% for routine dental extraction. However, its incidence is slightly higher in the case of mandibular third molar extractions (37.5% of cases) [12,13]. Most recommendations for the prevention of AO include disinfection of the mouth before, and after, extraction, the use of eugenol and antifibrinolytic agent-containing dressing, and systemic or topical post-extraction use of antibiotics [7].

AO treatment consists of pain management and encouragement of the healing process [4]. Hence, it alleviates pain, cleanses and disinfects the wound and temporarily obturates the wound to prevent further infection [14,15,16]. Systemic antibiotic use is contraindicated in any, but immunocompromised patients, to avoid unnecessary side effects and the threat of contribution to antibiotic strains resistance. Therefore, localized treatment is preferred [17]. Nowadays, frequent AO treatment protocol involves Alvogyl (Septodont, Inc, Wilmington, DE), a product which consists of eugenol and penghawar djambi, two substances with less than ideal properties (risk of allergic reaction, delayed healing process, non-resorbable, non-standardized composition, due to its natural origin). Eugenol serves as an analgesic and disinfectant. It is also associated with chronic inflammation, delay in wound healing, and risk of induction of allergic reaction. Penghawar djambi, natural mechanical styptic of plant origin, is composed of fibers, which tend to persist within the wound site and rarely even incorporate into the newly formed bone [18,19]. The pharmacological device tested in this study was designed to perform comparatively in pain alleviation and to promote healing, while avoiding the aforementioned side effects.

Hyaluronic acid (HA) is a non-sulfated glycosaminoglycan polymer, composed of disaccharide units (D-glucuronic acid and D-N-acetylglucosamine). Long chains of HA are the major components of synovial fluid, skin, mucosa, cartilage, and extracellular matrix, and assure tissue elasticity, support cell proliferation and migration, and serve as a lubricant. In tissue, damaged by trauma or infection, the HA long chains degrade and the resulting low molecular weight chains induce an inflammatory response, cell migration, and angiogenesis [20], which contribute to fetal healing (healing without scar creation) [21]. Extensively, the synthesis of HA increases during the first stage of healing [22], due to the effect of IL-8, TNF-α, and the presence of bacterial polysaccharides. This leads primarily to the activation of CD44-positive lymphocytes, and secondarily, to the induction of an inflammatory response. After the formation of granulation tissue, the role of HA changes as it starts to absorb free-radicals and so reduces oxidative stress in the new tissue [22].

Octenidine dihydrochloride (ODC) (C36H64Cl2N4) is non-specific cationic surface-active agent, that is widely used for antisepsis of skin or mucosa [23]. Due to its easy absorption and non-specific interaction with cell wall and wall membrane structure, ODC has wide antimicrobial, antifungal, and antiviral effects (enveloped viruses) [24]. In comparison to antibiotics, ODC with its non-specific mode of action is highly unlikely to aid the development of resistant pathogens. Moreover, according to Dogan et al. [25], ODC performs superior to other current oral cavity antiseptics. Its main advantages lay with the broad spectrum of pathogens it targets, its rapid onset of action, its lingering antimicrobial effect, its inability to induce microbial resistance or cause systemic side effects, and it is safe for children and pregnant women [25,26].

The aim of this clinical study was to evaluate the outcome of AO treatment by using a pharmacological device composed of ODC and HA. The device was designed to fulfill the following criteria: Disinfect the wound (provided by ODC), attach to the mucosa (HA), obturate the wound (HA), remain stable in the presence of saliva (HA), remain fully absorbable (ODC, HA), enhance the healing process (HA), remain non-allergic, and have analgesic effect (ODC, HA).

## 2. Materials and Methods

### 2.1. Composition of the Device

The sponge-like pharmacological device (Figure 1) is a lyophilized water solution of 2.5% HA, ODC and calcium chloride. The whole device weighed 35 mg, including 0.06 mg of ODC. The original HA molecular weight was 1.5 mDa, but during the lyophilisation process the molecular weight of the HA decreased to a range of 0.7–1.0 kDa. Calcium chloride promotes the ionic crosslinking between the carboxylic group and therefore slows the dissolving process. Therefore, the device is fully dissoluble, has antiseptic properties, and is malleable. The device was manufactured by Contipro a.s. (Dolni Dobrouc, Czech Republic).

### 2.2. Study Design

The clinical study was designed as a multi-center (8 centers in total), open-label, first-in-men study, and was approved by the ethical committee of the University Hospital Hradec Králové, ref. 201503 D02ZP, the Czech Republic State Institute for Drug Control ref. sukl122215/2015 an Clinicaltrial.gov ID: NCT04091399. Before they were included, patients were informed about the tested medical device and the alternative AO management, and signed the informed consent to participate in the study. The research staff was instructed not to discuss the anticipated treatment results with the patients to avoid a creation of expectations. This single cohort study does not feature a control group. Where necessary, a control group is represented by secondary data from comparative studies.

To be included in this study, all subjects had to be older than 18 years, able to fully understand and comply with the requirements of the study, and need to have been diagnosed with AO within 4 days prior to the introduction of the test-treatment. Excluded from the study were patients diagnosed with cancer, patients with a history of radiotherapy in the head and neck area, or those who have undergone bisphosphonate treatment within the last two years, patients who had been given antibiotics less than two weeks prior to AO onset, and patients with hypersensitivity or allergy to any substances contained in the tested device. In addition to the medical history criteria, the study’s subject pool excluded pregnant and lactating women, heavy smokers (more than 10 cigarettes per day), and people using illicit drugs. The study-subject selection criteria, clinical, and analytical methods mirrored previous research to allow comparison [27].

### 2.3. Treatment Protocol

Upon initiation of the study, the study subjects’ extraction wounds were examined and described by the medical professionals, and the subjects’ perceived-pain self-evaluation base data was recorded. The perceived pain was recorded on 0-100 mm Visual Analog Scale (VAS). Afterwards, the treatment was introduced. Firstly, the wound had been irrigated with 2 mL of 3% solution of H2O2 to disinfect the site and then flushed by 2 mL of Aqua pro injectione (Bieffe Medital, Capannori, Italy) to clear any remaining debris. Secondly, the tested device was applied into the extraction wound. This procedure was repeated on a daily basis for a maximum of 7 days, or until the pain subsided below 20 mm, and remained there for at least 2 days.

### 2.4. Study Objectives

The primary objective of this study was to evaluate the efficacy of the tested pharmaceutical device in the reduction of inflammation of the impacted alveolae, which was assessed subjectively by the study subjects who perceived the healing process through the lens of reduction in pain. The study subjects were asked to describe their perceived pain using the VAS.

### 2.5. Statistical Analyses

The statistical analyses performed, included both primary and secondary data, and both derived continuous and categorical variables.

Descriptive statistics were provided for each of the criteria using the following values; (1) for continuous data: Mean, standard deviation (SD), median, lower/upper quartile (Q1/Q3), minimum and maximum values, (2) for qualitative data: Absolute count (Count) and percentages (%). Hypotheses are tested at standard cutoff α = 0.05.

A descriptive analysis approach (including frequency tables) was used to assess clinical management, clinical outcomes, and healthcare resources used. When appropriate, the two-sided 95% confidence interval was obtained for population characteristics of a variable. All calculations and summaries were produced using R version 3.2.3 (R Core Team, Austria) [28].

## 3. Results

### 3.1. Patient Cohort

This study included 58 study subjects (35 females, 23 males) in average age 36.1 ± 12.2 years. Patient distribution by medical center is listed in Table 1. Thirty-six patients underwent an extraction of lower 3rd molar, 9 patients underwent extraction of 1st or 2nd lower molar, 3 patients underwent extraction of 1st or 2nd upper molar, 3 patients underwent extraction of 1st or 2nd lower premolar, 3 patients underwent extraction of 1st or 2nd upper premolar, 2 patients underwent extraction of upper 3rd molar, and 2 patients underwent extraction of lower canine. Forty-five patients underwent the standard extraction of teeth, while 13 underwent complicated extraction with mucoperiosteal flap flip. Ten patients admitted smoking equal, or less than 10 cigarettes per day, while 48 patients were non-smokers (Figure 2). From a total of 58 study subjects, there were 8 patients who did not complete their treatment due to a loss of contact, therefore all results are calculated from a total of 50 subjects who completed the study.

### 3.2. Study Objectives

After administration, the tested device started to change from sponge form into an expanding gel (Figure 3). The gel layer showed a high adhesion to wet tissue surfaces and helped keep the device in situ. Moreover, it prohibited the saliva from getting closer into the core of the tested device, and due to this, prolonged the time until the device was fully dissolved.

Most of the patients described pain relief directly after administration, even though this device does not contain any anesthetics. After 18 h the pain aggravated again but never reached the previous intensity. The tested product did not stay within the extraction wound for more than 24 h. According to the patients ’ description, this device fully dissolved after about 16–20 h.

From a total of 50 study subjects who finished the study, 48 have reported that their perceived AO pain subsided below 20 mm VAS, within 7 or less administrations treatment-days, or of the tested pharmacological device, which eventually represented the success rate of 96% (95% confidence interval of 75.75% to 97.80%) (Table 2). Two study subjects reported continuous pain above 20 mm in the VAS, meaning 40 mm, and 28 mm respectively, even after 7 administrations of the tested device. Therefore, their treatment is considered unsuccessful with respect to this study.

The extent of exposure, until the VAS value fell under 20 mm, was from 1 to 9 days, the mean value of 4.8 days, and a median of 4 days. On average, the pain fell under VAS 20 mm within 4 administrations at 40 of 50 patients, which represents 80% of patients.

In time, the changing pain level was assessed as the change in mm of VAS between baseline visit and the final visit. Mean VAS value was reduced from 65.5 mm on baseline visit to 6.1 mm on the final evaluation visit (Figure 4), with a difference 59.4 mm (95% confidence interval of 53.5 to 65.4 mm). This change is significant with p-value < 0.001 (with paired T-test used).

There were no adverse effects of the tested device detected in any of the patients included in the study.

## 4. Discussion

The AO therapeutic procedure is mostly based on pain alleviation, obturation, and disinfection of the extraction site. The majority of protocols and devices are based on clinical experiences; not supported by research. Even more, some devices are based on natural origin compounds, where the effective substances are not identified, and therapeutic effects are not fully understood. Many protocols involved substances, where usage is already forbidden for general medicine [29].

Ideal devices should be fully absorbable, provide coverage of the extraction socket, eradicate the microbial infection, relieve pain, and induce the healing process. Commonly available devices are based on eugenol and/or antibiotics. Eugenol-based devices do not support healing, and according to some studies, they even prolong it due to the adverse effects of eugenol on the damaged tissue [29]. The presence of a high dose of antibiotics is controversial. Even though antibiotics provide good antimicrobial effects within the extraction wound, they are slowly released into the saliva and swallowed by the patient. This leads to the absorption of a low dosage of the antibiotics; their concentration in blood does not achieve minimal inhibitory concentrations, and leads to the development of antimicrobial resistant pathogens [4]. On the other hand, most of the magistral devices used substances that are toxic and already forbidden for use in general medicine [29].

Taberner-Vallverdú et al. [30], in a systematic review, analyzed the effectiveness of eight different methods for the management of dry socket (Zinc oxide eugenol (ZOE), Alvogyl (Septodont, Cambridge, Canada), G.E.C.B. Pastille (Sultan Company, Kuwait, Kuwait), Vitamin C, SaliCept Patch (Carrington Laboratory, Irving, USA), plasma-rich in growth factors (PRGF), topical anesthetic gel Oraqix (Dentsply Pharmaceutical, York, USA), and low-level laser therapy (LLLT, Lambda Laser Products, Vicenza, Italy)). The effectivity of treatment approaches were compared using two variables: pain alleviation and extraction wound healing. According to 48 h values from treatment initiation, topical anesthetic gel is more effective than eugenol [31]. Kaya et al. [32] compared the effectiveness of Alvogyl, SaliCept and LLLT in pain reduction and concluded that LLLT performed superiorly to SaliCept and Alvogyl.

This single cohort study tests new pharmacological device, based on a combination of lyophilized hyaluronic acid (HA) and octenidine dihydrochloride (ODC), in the treatment of alveolar osteitis. The HA provides obturation of the extraction socket, induces the healing process, and serves as a carrier for octenidine, which provides the antimicrobial effect.

There are two major aspects of how to evaluate the effectiveness of AO treatment—wound healing and pain relief. Due to the content of hyaluronic acid in newly tested devices, the acceleration in wound healing was confirmed in comparison to frequentlyused treatments, using Alvogyl. Regarding the different therapies compared in the study by Kaya et al. [32], none of the patients treated with LLLT therapy had an empty socket after three days of application, so all had begun the healing process, while in the group treated with Alvogyl, little more than half of the patients had started the healing process.

In the majority of the study, subjects reported pain subsidy below 20 mm VAS in a span of four days of treatment. The prolonged healing until more than the average four administrations in eight patients, or the failure of the treatment at two patients, correlates with the predominant factors. Out of 10 patients, three patients admitted smoking even during the clinical study, five of them were pretreated with Alvogyl before they were enrolled in this study, and three of them underwent the extraction despite the acute inflammation in the site. One patient reported both, smoking during the study, and pretreatment with Alvogyl. This correlates with the theory that smoking slows down the healing process [3,7]. Therefore, it represents a risk factor for the AO incidence and treatment. Eugenol contained within the Alvogyl helps with management of the pain and disinfects the wound. However, it causes chronic inflammation which results in delayed healing [29].

There were no adverse device effects detected during the course of the study.

Our newly developed medical formula, based on hyaluronic acid and octenidine dihydrochloride, shows very good results in terms of efficacy, and it is fully absorbable. On the other hand, due to the low stability in the presence of saliva, the treatment protocol needs to be repeated every day, while some other AO-management protocols offer less frequent administrations, e.g., Alvogyl as per a study by Kaya et al. [32] can be replaced once every 2–3 days.

The main weakness of the presented study is the lack of control group. The reason for this is that nowadays, there is no treatment of AO which can be taken as a gold standard. The most frequent therapy includes pain management and the encouragement of healing process. Therefore, the aim of the study was not to compare different treatment protocols, but the main goal of our study was to conclude that there are no safety concerns in the use of a pharmacological device, composed of ODC and HA in treatment of AO, and to prove that the device fulfills the criteria sought in the AO treatment.

This weakness was partially remedied by the incorporation of secondary data provided in the aforementioned study by Kaya [32]. According to Kaya, before administration of Alvogyl the dry socket was first curetted and irrigated. The pain disappeared in five out of 26 patients (19.2%) after three days. After seven days, the pain disappeared in 22 out of 26 patients (84.6%). In comparison, after the third application (fourth visit), 31 out of 50 patients were considered healed (which represents 62%) and at day seven, 48 out of total 50 patients were considered healed (96%) in our study.

In the future, we will focus on further research featuring the newly created medical device, by comparing its benefits with commonly AO management protocols used.

## 5. Conclusions

According to the acquired data, there is no safety concern, which would prevent a wide use of the new device. The results of the study propose that lyophilized HA, combined with ODC and can be used for the treatment of AO. Except for long-term stability, this device fulfills all proposed criteria — the ability to disinfect a wound, attach to the mucosa, obturate a wound, and complete absorption, analgesic, and non-allergic effects.

## 6. Patents

The medical device is protected by utility model Hyaluronan and octenidine dihydrochloride-based dental composition, CZ 28634 U1.

## Figures and Tables

**Figure 1 ijerph-16-03698-f001:**
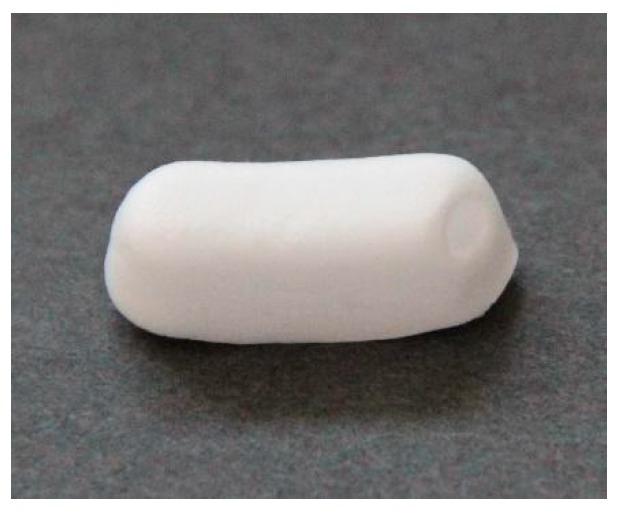
The tested medical device. The size of the medical device is 2 × 0.5 cm on the widest site. The sponge like structure allows to perform the device according to the specific shape of the socket.

**Figure 2 ijerph-16-03698-f002:**
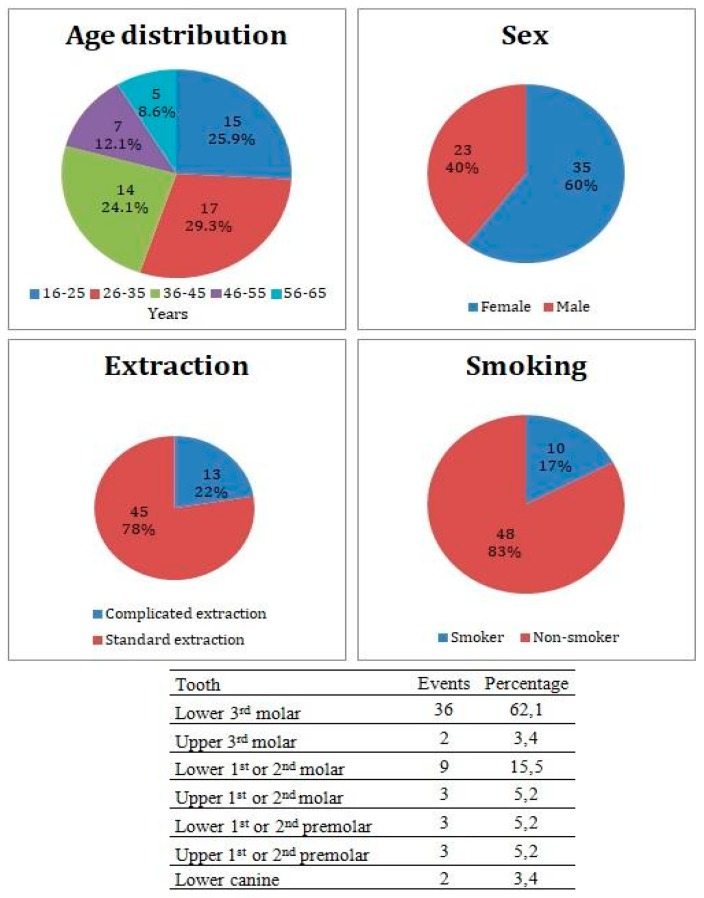
Patients cohort characteristics and incidence of the Alveolar Osteitis (AO) events divided by the extracted tooth.

**Figure 3 ijerph-16-03698-f003:**
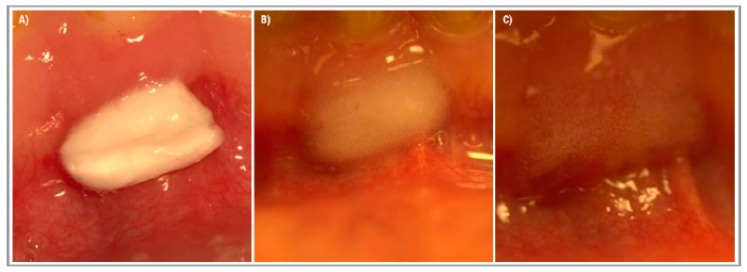
The effect of saliva on the tested medical device applied to the gums in the lower frontal area. The applied medical device was cut to the size 1 × 0.5 cm. (**A**) Right after administration, (**B**) 2 h after administration, and (**C**) 4 h after administration.

**Figure 4 ijerph-16-03698-f004:**
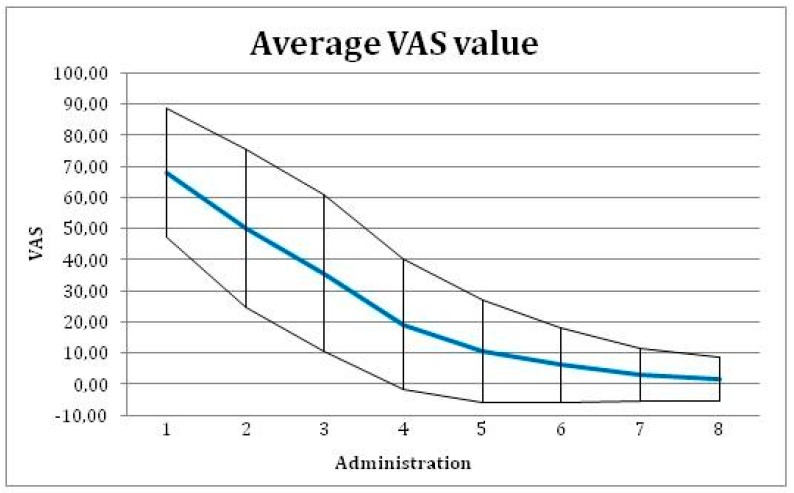
Average Visual Analog Scale (VAS) value with ± SD for each administration of the tested medical device.

**Table 1 ijerph-16-03698-t001:** Numbers of treated patients divided by the centers.

Center	Number of Treated Patients
1	22
2	4
3	1
4	7
5	7
6	4
7	5
8	8

**Table 2 ijerph-16-03698-t002:** Numbers of patients with successful reduction of inflammatory symptoms divided by each administration.

Device Administrations	Number of Observed Events	Proportion of Fade Away of Inflammatory Symptoms (%)	95 % Confidence Interval
1	10	20.0	6.9, 26.4
2	9	38.0	19.5, 43.8
3	12	62.0	39.5, 65.9
4	9	80.0	56.7, 81.4
5	5	90.0	68.1, 90.4
6	1	92.0	70.7, 92.8
7	2	96.0	75.7, 97.8

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
