# Peer review of "Hyaluronic Acid-Based Medical Device for Treatment of Alveolar Osteitis—Clinical Studyâ€"

_ijerph, 2019, doi:10.3390/ijerph16193698_

Round 1
Reviewer 1 Report
The authors addressed all queries, thank you.
Author Response
Dear respectable reviewer,
at the beginning of mine response please let me thank you on the behalf of all authors for your time and valuable revision.
Sincerely,
MUDr. Jakub Suchánek, Ph.D.
Reviewer 2 Report
This paper deals with a clinical study in which the evaluation of a hyaluronic acid patch loaded with Octenidine dihydrochloride is carried out for Alveolar Osteitis treatment.
More details are required in the description of the device. Indeed the type of “device” should be specified.
The concentration of hyaluronic acid solution and the loading of Octenidine dihydrochloride should be specified. Taking into account the importance of the molecular weight of hyaluronic acid, as authors claim, it must be specified.
Line 20: which are the fully absorbable medicaments?.
Line 39: typographical mistake.
Line 54: “with less than ideal properties” Which are those disadvantages?
What is the role of CaCl2 stabilizazing? Does it promote an ionic crosslinking according to the carboxylic groups of the polymer, like in the case of alginate?
Mucoadhesiveness of the devices is related to its dissolving process, thus degradation kinetic of the device should be evaluated previously in vitro in order to can explain in vivo results.
Author Response
Dear respectable reviewer,
at the beginning of mine response please let me thank you on behalf of all authors for your time and valuable comments. We did our best to explain them or improve the manuscript according to them.
More details are required in the description of the device. Indeed the type of “device” should be specified. The description was added, even though we are not sure what the reviewer thought by “type of device”. The concentration of hyaluronic acid solution and the loading of Octenidine dihydrochloride should be specified. Taking into account the importance of the molecular weight of hyaluronic acid, as the authors claim, it must be specified. The sponge-like pharmacological device (Figure 1) is a lyophilized water solution of ODC, 1.5 mDa HA, stabilized with calcium chloride by creation of ionic crosslink between the carboxylic groups. During the lyophilisation process the molecular weight of the HA decrease at the 0.7 – 0.8 kDa. As such it is fully dissoluble, has antiseptic properties and is malleable. Line 20: which are the fully absorbable medicaments?. The tested device is a sponge-like material composed solely of fully dissoluble medicaments (hyaluronic acid, calcium chloride and octenidine dihydrochloride). Line 39: typographical mistake. This part was rewrittenAmong the most common causes and risk factors of AO are trauma or complicated extraction, smoking, or acute inflammation of periodontal tissues (e.g. dentitio difficilis) prior to extraction. Also, the use of oral contraception leads to 10 times higher likelihood of AO. (Napsala bych to asi takhle, Nela)
Line 54: “with less than ideal properties” Which are those disadvantages? risk of allergic reaction, delayed healing process, nonresorbable, non standardized composition due to the natural origin – added to manuscript What is the role of CaCl2 stabilizing? Does it promote an ionic crosslinking according to the carboxylic groups of the polymer, like in the case of alginate? As the reviewer mentioned, the role of CaCl2 is decrease the speed of dissolving by creation of crosslink between the carboxylic groups. – added to manuscript Mucoadhesiveness of the devices is related to its dissolving process, thus degradation kinetic of the device should be evaluated previously in vitro in order to explain in vivo results. All degradation kinetics was tested repeatedly under in vitro condition during optimization of the HA molecular weight and concentration of the ODC and CaCl2. – not added to the manuscript
Sincerely,
MUDR. Jakub Suchánek, Ph.D.
This manuscript is a resubmission of an earlier submission. The following is a list of the peer review reports and author responses from that submission.
Round 1
Reviewer 1 Report
This study regards the use of new pharmacological agents for treatment of alveolar osteitis. The study has several severe methodological flaws due to which it is impossible to recommend it for publication.
1. The authors keep calling their study a clinical trial. This is not a clinical trial, but a clinical study. In any case it does not seem to be registered with clinicaltrials.gov. It is an absolute requirement to register studies with some internationally recognised clinical body. Having ethical approval alone is not enough. All clinical studies must be registered. there is no recognised journal in the world that will publish a study unless it is registered with a registration body.
2. The most fatal flaw in this study is lack of a control group. It is a real shame that the authors did not use a control group as there are several agents available currently for the management of AO. There can be no justification in the current scientific environment for not including a control, group in this study. The authors keep calling the study a clinical trial despite this and there can be no clinical trial without at least two interventions or one intervention plus control. For this reason alone it would be completely unjustifiable to publish this study.
3. The manuscript needs to be completely rewritten and authors should seek help with how to write a manuscript for publication.
Given the fatal flaws it cannot be recommended for publication
The authors must be advised to seek advice on conduct of clinical studies for the future.
Author Response
Reviewer 1:
Dear respectable reviewer,
at the beginning of mine response please let me thank you on the behalf of all authors for your time and valuable comments. We did our best to explain them or improve the manuscript according to them.
1) The authors keep calling their study a clinical trial. This is not a clinical trial, but a clinical study. In any case it does not seem to be registered with clinicaltrials.gov. It is an absolute requirement to register studies with some internationally recognised clinical body. Having ethical approval alone is not enough. All clinical studies must be registered. there is no recognised journal in the world that will publish a study unless it is registered with a registration body.
According to this comment, we corrected all terms of clinical study to single cohort study. This term suits precisely the description of this manuscript. We have proceeded the registration of our study into clinicaltrials.gov database. Our request is under review and it is supposed to be completed in the upcoming 7 business days. After that, we will be able to fully obligate all requests mentioned in this comment.
2) The most fatal flaw in this study is lack of a control group. It is a real shame that the authors did not use a control group as there are several agents available currently for the management of AO. There can be no justification in the current scientific environment for not including a control, group in this study. The authors keep calling the study a clinical trial despite this and there can be no clinical trial without at least two interventions or one intervention plus control. For this reason alone, it would be completely unjustifiable to publish this study.
Nowadays, there is no treatment of Alveolitis sicca (AO) which can be taken as a gold standard. The most frequent therapy includes pain management and encouragement of healing process. Therefore, the aim of the study was not to compare different treatment protocols, but the main goal of our study was to conclude that there are no safety concerns in the use of a pharmacological device composed of ODC and HA in treatment of AO and prove that the device fulfills the criteria that are sought in the AO treatment: to disinfect the wound, attach to the mucosa, obturate the wound, be stable in the presence of saliva, be fully absorbable, enhance the healing process, be non-allergic and have analgesic effect. .
3) The manuscript needs to be completely rewritten and authors should seek help with how to write a manuscript for publication.
We rewrote the content of the manuscript according to requirement of another reviewers. Now, it depicts precisely how the study was performed.

Reviewer 2 Report
Dear authors , the manuscript deals with a very interesting subject. However the manuscript must be improved:
This is not a trial. This is a single cohort study. Please remove all mentions to trial and replace them with the appropriate study designation.
Line 168: this belongs in the Discussion section as you are comparing the results to another study. Please move them.
Line 208-211: this represents a description of the rssults and should be moved to the Results section.
Line 221-223: Please discuss the results with Kaya et al. But without calling it a control group.
Discussion section: Needs to be improved. Please address the results using other studies on the evaluation of different products and their potential effect or a more developed discussion of the characteristics of hialuronic acid and octenidine supported by scientific studies.
Line 236-231: the conclusions need to be adapted as this is not a clinical trial. Also on the Discussion section it should be inserted a sentence of study strengths, study weakenesses and suggestions for future research (where you should mention the need of climical trials evaluatong this product with Alvogyl for example)
Author Response
Reviewer 2:
Dear respectable reviewer,
at the beginning of mine response please let me thank you on the behalf of all authors for your time and valuable comments. We did our best to explain them or improve the manuscript according to them.
1) This is not a trial. This is a single cohort study. Please remove all mentions to trial and replace them with the appropriate study designation.
The manuscript was corrected according to this suggestion
2) Line 168: this belongs in the Discussion section as you are comparing the results to another study. Please move them.
The manuscript was corrected according to this suggestion
3) Line 208-211: this represents a description of the results and should be moved to the Results section.
The manuscript was corrected according to this suggestion
4) Line 221-223: Please discuss the results with Kaya et al. But without calling it a control group.
The manuscript was corrected according to this suggestion
5) Discussion section: Needs to be improved. Please address the results using other studies on the evaluation of different products and their potential effect or a more developed discussion of the characteristics of hyaluronic acid and octenidine supported by scientific studies. According to this comment, we added the results of other studies using different approaches in AO treatment. The main properties of hyaluronic acid and octenidine are mentioned in previous part of the manuscript.
6) Line 236-231: the conclusions need to be adapted as this is not a clinical trial.
The manuscript was corrected according to this suggestion
7) Also on the Discussion section it should be inserted a sentence of study strengths, weaknesses and suggestions for future research (where you should mention the need of clinical trials evaluating this product with Alvogyl for example)
The manuscript was corrected according to this suggestion

Round 2
Reviewer 2 Report
Dear authors,
Thank you for your changes performed to the manuscript. Nevertheless there are some minor changes this Reivewer deems necessary in order to make the manuscript clearer.
I attach a pdf with my comments directly on the text. Please also upload the figures to be evaluated as there are no figures in the manuscript but you referenced 3 figures.

Author Response
Reviewer 2:
Dear respectable reviewer,
at the beginning of mine response please let me once again thank you on the behalf of all authors for your time and valuable comments. We went through your comments and the manuscript was corrected according to them.
Sincerely,
MUDR. Jakub Suchánek, Ph.D.
